# Technologies Enabling Single-Molecule Super-Resolution Imaging of mRNA

**DOI:** 10.3390/cells11193079

**Published:** 2022-09-30

**Authors:** Mark Tingey, Steven J. Schnell, Wenlan Yu, Jason Saredy, Samuel Junod, Dhrumil Patel, Abdullah A. Alkurdi, Weidong Yang

**Affiliations:** Department of Biology, Temple University, Philadelphia, PA 19122, USA

**Keywords:** mRNA, single-molecule super-resolution microscopy, SMLM, MS2-MCP, FISH, smFISH, seqFISH, MTRIPs, CRISPR-Cas9, CRISPR-Cas13, molecular beacons

## Abstract

The transient nature of RNA has rendered it one of the more difficult biological targets for imaging. This difficulty stems both from the physical properties of RNA as well as the temporal constraints associated therewith. These concerns are further complicated by the difficulty in imaging endogenous RNA within a cell that has been transfected with a target sequence. These concerns, combined with traditional concerns associated with super-resolution light microscopy has made the imaging of this critical target difficult. Recent advances have provided researchers the tools to image endogenous RNA in live cells at both the cellular and single-molecule level. Here, we review techniques used for labeling and imaging RNA with special emphases on various labeling methods and a virtual 3D super-resolution imaging technique.

## 1. Introduction

### 1.1. Why Image RNA?

The flow of genetic information is a multi-stage process centered around RNA metabolism. Since Francis Crick proposed the central dogma of molecular biology [1], RNA has been a central focus in the field of molecular and cellular biology. RNA is a multi-functional macromolecule that plays an essential role in gene expression and regulation. In gene expression, messenger RNA (mRNA) acts as templates that carry the genetic information from the DNA blueprint to ribosomes. Transfer RNA (tRNA) acts as an amino acid transporter that helps decode an mRNA sequence into a protein. Ribosomal RNA (rRNA), one of the main components of the ribosome, has shown deep involvement in ribosomal subunit association, tRNA binding, and translocation during translation [2,3]. In gene regulation, an overwhelming amount of evidence has demonstrated that small regulatory RNA is associated with cellular regulation via various mechanisms. For example, small interfering RNA (siRNA) [4] and small hairpin RNA (shRNA) [5,6] regulate gene expression by a phenomenon known as RNA interference (RNAi), whereas microRNA (miRNA) is involved in translational activation and repression [7].

Interestingly, the *RNA world* hypothesis has drawn more attention recently in the scientific world with emerging evidence [8,9] suggesting that life arose from self-replicating RNA-based genes and only later did organisms develop the ability to store genetic information in more stable DNA [10]. An exciting observation supporting this hypothesis is RNA enzyme, which was found undergoing self-sustained replication independently from proteins or other biological materials [11]. Recently, a new study suggested that RNA might be the key to the initial peptide synthesis [9], an observation that provides significant support for the *RNA world* hypothesis as a model for the origins of life. In addition to enhancing our knowledge of cellular activities and revealing the possible origins of life, RNA also has been demonstrated with promising disease diagnostic and therapeutic potentials. RNA vaccines against SARS-CoV-2 have exhibited high prophylactic efficiency against the disease during the two years since their development [12,13]. Further, several therapeutics based on RNAi have been developed. The first of these therapeutic agents were approved by the US Food and Drug Administration in 2018, with many more on the horizon [14]. Lastly, the use of extracellular RNAs in biofluids has rapidly grown in diagnostic research as a non-invasive method of monitoring disease [15]. Together, the detailed advancements listed above highlight the significance of RNA in the field of biology and medicine; Thus, comprehensive studies surrounding this macromolecule are needed.

### 1.2. RNA Localization and Imaging: Seeing Is Believing

Controlling the localization of RNA is a widespread, evolutionarily conserved, and efficient way to target gene products to a specific region of a cell or embryo [16]. Using mRNA as an example, after the initial transcription, mRNA proceeds through post-transcriptional modification such as alternative splice, nucleocytoplasmic transportation across the nuclear pore complex, localization to the ribosome, translation, and finally degradation. These steps are highly coordinated and tightly regulated both spatially and temporally. Unsurprisingly, the subcellular localization of RNA has been determined to be one of the fundamental mechanisms of cell polarization. One good example is β-actin mRNA expression in moving fibroblast cells, where mRNAs for β-actin are concentrated at the moving edge. This subcellular preference was believed to produce a localized high protein concentration that would facilitate directional motility for the cell [17]. At the same time, mRNA localization has been linked to embryonic development and patterning. For example, the *Drosophila melanogaster* gene *bicoid* functions as an anterior body pattern organizer. The *bicoid* mRNA is localized only in the anterior pole region of the egg via random dynein-mediated transport and anchoring [18]. After fertilization and translation of the transcript, the protein product named *bicoid* spreads posteriorly, generating a concentration gradient for arterial/posterior patterning.

The examples above show that mRNA function is closely tied to its subcellular localization and transportation, and thus methods and tools are needed to examine this aspect of RNA research. Unfortunately, conventional biochemistry and genetic tools primarily focus on identifying the properties of the RNA molecule, such as length, sequence, structure, and functions, whereas pulldown assays provide insight into the protein-RNA interactions. In order to obtain a complete spatial-temporal profile of RNA throughout its entire lifespan, from transcription to degradation, methods to visualize RNA within cells are required. Such methods are critically important to enhance our understanding of RNA and thus offer unparalleled opportunities for advancement in cellular and molecular biology, therapeutic discovery, and medical diagnostic.

### 1.3. Difficulties Associated with Imaging mRNA

Generally speaking, the size of mRNA is far smaller than the resolution limit of conventional light microscopy. Thus, only two approaches are feasible regarding mRNA imaging: electron microscopy (EM) and fluorescent light microscopy. Among various EM techniques, two techniques have been used in mRNA imaging: first, an EM-level adaptation of in situ hybridization (ISH) technique that combines antisense probes and gold-coupled antibodies for detection [19,20]; and second, Cryo-EM, a mainstream technology in structural biology for architectural study [21,22]. The EM approach for mRNA imaging provides excellent spatial resolution, structural, and ultrastructural information. However, it has limitations, including the lack of temporal resolution due to the use of fixed samples, overly complicated sample preparation, artifact susceptibility [23], and low labeling efficiency [24]. EM-ISH is still a commonly used technique for obtaining images with a high spatial resolution that reveal the cellular distribution of mRNA in fixed tissue, whereas Cryo-EM is the predominant method for mRNA structure. Observation of mRNA in live tissue is required to explore mRNA transport dynamics, maintenance and regulatory mechanisms, and localization.

Another challenge in the fluorescent light microscopy approach of mRNA imaging is how to label the RNA. Fluorescent light microscopy relies mainly on fluorophores, including fluorescent proteins and dyes. At the time of this writing, no naturally fluorescent RNA has been discovered [25]; as a result, extensive efforts have been dedicated to developing fluorescence labeling strategies for mRNA, which will be discussed in detail later in this review. 

An appropriate fluorophore used to label RNA needs to have three outstanding characteristics: quantum yield, extinction coefficient [26], and photostability. The relative brightness of a fluorophore is dependent upon the quantum yield and extinction coefficient of the fluorophore, causing some fluorophores to emit more light than others under the same excitation power. Thus, it can provide better spatial and temporal resolution without risking cell viability in live cell imaging. At the same time, fluorophore stability is associated with its resistance to photobleaching. Photobleaching is a dynamic process in which a fluorophore exposed to excitation light undergoes photoinduced chemical destruction, thus losing its ability to fluoresce [27]. A fluorophore with higher stability will stay unphotobleached for an extended time, allowing for more prolonged imaging. Recent developments in improved sensitivity in imaging systems and dyes with improved quantum yield and stability provide an opportunity for long-term RNA imaging, making imaging of the whole RNA lifecycle achievable. The use of quantum dot–based nanobeacons for mRNA labeling and imaging has been reported in recent publications [28,29], suggesting a promising future in real-time live-cell RNA imaging using newly developed fluorophores.

Lastly, choosing an imaging method that matches the experimental approach can be challenging but essential. The experimental design must take into consideration the speed of image acquisition, temporal resolution, signal-to-noise ratio, and effects on cell viability. Wide-field fluorescent microscopy with deconvolution could provide sufficient speed and sensitivity to study mRNA dynamics [30]. However, this method suffers from photon collection and spatial resolution deficiencies. Laser scanning confocal microscopy (LSCM) is better than widefield at removing out-of-focus light and improving resolution. Unfortunately, the scanning approach inherent to LSCM is usually inappropriate for imaging highly dynamic processes because of the longer image acquisition time inherent in the technique. 

In the past two decades, a series of revolutionary techniques termed *super-resolution microscopy* (SRM) has been developed that bypass the diffraction limit. The diffraction limit is a barrier in optical microscopy caused by the physical property of light, restricting the optical resolution to roughly 250 nm [31]. There are generally two subgroups of SRM [32], the first being SRM by single-molecule-localization-based imaging such as stochastic optical reconstruction microscopy (STORM) [33] and photoactivated localization microscopy (PALM) [34] and the second being SRM by spatially patterned excitation such as stimulated emission depletion (STED) microscopy [35] and structured illumination microscopy (SIM) [36]. STED and SIM achieve super-resolution using patterned illumination to differentially modulate the fluorescence emission of molecules within the diffraction-limited volume [37]. Thus, those techniques have been well used with fixed samples labeled with ISH and immunofluorescence to obtain sub-diffraction images [38,39,40]. On the other hand, SRM by single-molecule-localization-based imaging, enables the determination of the localization of a single fluorophore. Currently, research involving trajectory and single molecule tracking of mRNA is primarily performed using single-molecule localization microscopy (SMLM) techniques in live samples [41,42,43].

### 1.4. Single-Molecule Super-Resolution Imaging

SMLM methods typically utilize conventional wide-field excitation and achieve super-resolution by fitting and localizing individual molecules, which are subsequently utilized to form a complete image in a pointillistic fashion [43]. Since the inception of these techniques, they have become broadly adopted in life science research because of their superior spatial resolution, which in most cases can achieve 20 nm lateral and 50 nm axial resolution or better [31]. SMLM is fundamentally based on the fact that the spatial coordinates of single fluorescent molecules can be determined with high precision from an isolated point spread function (PSF). The single PSF can be approximated with a Gaussian intensity distribution, allowing the exact center of the corresponding single emitter to be determined, even if it sits between two pixels of the imaging system [44]. In order to avoid overlapping between PSFs, fluorescent emissions of distinct molecules have to be separate. There are other ways to ensure a temporal separation between PSFs; the most commonly used approach exploits photoswitchable or photoactivation probes. Supposing the majority of fluorophores in a sample are converted to a dark state, and only a tiny subset of the population switches back on, the probability of two emitters residing near each other will be minimal. Under these conditions, one can calculate and record the location of each emitter in this subset. After bleaching or switching off the current emissive fluorophores, a new subset can be activated. This process can be repeated multiple times. When a sufficient number of location data are accumulated, the structure associated with the fluorophores can be reconstructed from hundreds of subsets of emitter distribution. It is also important to note that when referring to the single-molecule localization of RNA, it is understood that a single sequence of RNA is intended, regardless of how many fluorescent labels are present. Multiple fluorescent labels attached to a single sequence creates additive fluorescence and generates a desirable signal-to-noise ratio resulting in a better single-molecule localization of the target sequence. 

### 1.5. Utilizing Single-Molecule Super-Resolution Imaging for mRNA

Before diving into various SMLM techniques used to image mRNA, the advantages and disadvantages of live cell imaging and fixed samples must first be addressed. Fluorescence microscopy of living or fixed cells is entirely dependent upon suitable labeling and detection strategies, as well as the overall scope of the study. In fixed cells, because all cellular activities and movements have been terminated during fixation, this imaging strategy allows repeat capture to maximize localization precisions, which contributes to the spatial resolution advantages of STORM and PALM. However, cell fixation also has significant limitations. For instance, fixed cells are no longer capable of providing dynamic information, and exhibit changes in cellular structures or molecules caused by the fixation process [23].

Live cell imaging is a closer representation to the natural state of the cell. Whereas fixed cells are best described as in situ, live cells remain our best method of obtaining data regarding the behavior of cells in their native environments. For example, live cell imaging enables researchers to obtain real-time measurements at the temporal frequency necessary to sample the dynamics of most biological processes adequately [45]. However, the cell viability associated with photobleaching and phototoxicity should be considered carefully, as live cells are susceptible to photodamage that impacts the behavior and viability of the cell, thereby rendering any data acquired from that cell suspect. At the same time, as particles freely move in three-dimensional space within live cells, prolonged single particle tracking can be challenging, often requiring significant datasets to observe dynamic processes [46].

Another consideration when comparing the two primary super-resolution approaches is the size of the dataset required to generate super-resolution localizations. Depending on the samples and applications involved, SMLM methodologies often require thousands of frames to reconstruct a high-quality image from an individual localization event, sometimes requiring minutes or hours to capture. As a result, many SMLM techniques are often performed on chemically fixed cells to prevent cell movement and facilitate the localization of single molecules. Despite this consideration, SMLM techniques have enabled significant progress in the arena of imaging mRNA dynamics and mRNA-protein interactions. For example, SRM and SMLM techniques have been utilized to great effect to localize a single RNA macromolecule with high spatial resolution, providing information regarding RNA binding protein (RBP) partners. However, it should be noted that not all SMLM techniques will achieve molecular resolution. In a study published in 2017 [47], an SMLM-based method was developed to quantitatively analyze RNA-protein interactions with sub-diffraction resolution. Using this method, the authors suggested that many known RBPs from traditional biochemical pulldown assays did not necessarily bind the mRNA in situ. Further, the authors suggested that as a result of this deficiency in biochemical pulldown methodologies to fully encapsulate the interactome of in vivo RNA, super-resolution imaging methodology must be paired with biochemical methodologies to further interrogate the interacting proteins associated with RNA in vivo.

RNA has also been studied utilizing the single-particle tracking capacity of SMLM methodologies. The principles of SMLM can be expanded to track single-particles within live cells where a labeled single RNA macromolecule is localized with nanometer precision and observed over a period of time. Such fine-detailed and dynamic information, when paired with structural information of the cell, provides invaluable information pertaining to the dynamics and transport behaviors of RNA. A distinct advantage of tracking single particles is the ability to derive 3D information regarding the dynamic behaviors of RNA. One way in which this has been utilized is through the use of single-particle localization in two-dimensions to derive virtual three-dimensional (3D) information using a computational algorithm via single-point edge-excitation sub-diffraction (SPEED) microscopy. SPEED microscopy has been used to track messenger ribonucleoprotein (mRNP) movement through the nuclear pore complex (NPC) of eukaryotic cells [41,48,49]. This technique is specifically designed to track and record 2D spatial locations of fast-moving fluorophores within a rotationally symmetric biological structure with a spatiotemporal resolution of approximately 10 nm and 0.4 ms, respectively [41,48,49]. After image acquisition, post-localization 2D-to-3D transformation is applied to obtain 3D super-resolution structural and dynamic information. Besides the computational 2D-to-3D algorithm specially designed for rotational symmetric biological channels in the nuclear pore, another SMLM approach combined with multifocus microscopy (MFM) was also developed within the past few years. The MFM method produces focal stacks of high-resolution 2D images simultaneously displayed on a single camera [50]. It is based on using a diffractive grating to form multiple focus-shifted images, thereby enabling multiple captures on different z-axis positions without sacrificing temporal resolution. Unfortunately, the use of MFM would also decrease the light efficiency to ~60% compared with the wide-field microscope. Maximizing the light efficiency is critical for the localization precision for the SMLM. As a result, only a few publications used MFM for RNA localization, and one of them showed a sub-80 nm localization precision [51].

Taken together, super-resolution microscopy methodologies present an attractive option for obtaining both dynamic and static information regarding RNA localization, lifecycle, and interacting proteins. Detailed within this manuscript is a series of techniques developed to facilitate the imaging of RNA, with mRNA being given special consideration.

## 2. RNA Imaging Methods

### 2.1. FISH

Fluorescence in situ hybridization (FISH) is a valuable method for imaging nucleic acids. First developed by Bauman and colleagues in 1980 [52], this method was originally intended to be a way to replace and improve on old methods of in situ hybridization (ISH) that utilized ^3^H- or ^125^I-labeled radioactive hybridization probes. By combining the hybridization approach in the previous autoradiography methods with observations made by Rudkin & Stollar [53], wherein it was demonstrated that RNA:DNA hybrids could be targeted via a fluorescently labeled antibody, Bauman and colleagues devised a mechanism to specifically target a nucleic acid by introducing a complimentary RNA sequence covalently bound to a fluorescent probe that would then hybridize with the desired sequence [52] (Figure 1A).

The early incarnation of FISH was shown to facilitate imaging of mitochondrial DNA, viral DNA within human tissue culture cells, and 5S rRNA [52]. Whereas this technique demonstrated the theoretical possibility of FISH to image hybridized RNA, the first true mRNA FISH was performed by Singer & Ward in 1982 to visualize actin mRNA in chicken skeletal muscle culture [54]. This visualization was accomplished using a DNA probe labeled with biotin. After the hybridization of DNA to RNA, a goat anti-biotin primary antibody followed by a rabbit derived anti-goat secondary antibody conjugated to rhodamine was added [54] (Figure 1B). This complex showed higher fluorescence than the direct comparison approach originally devised by Bauman and colleagues [55]; however, the size of the complex made it unwieldy. One significant drawback to this approach is the relative difficulty of differentiation between signal and noise. There exists the possibility that probes not associated with their target could generate background noise, thereby rendering quantitative analysis of mRNA with this methodology extremely difficult.

In response to these difficulties, ISH methodology has been improved over the intervening years, primarily focusing on two areas: the development of new and novel fluorophores and the addition of multiple fluorophores to a single probe. The original incarnation of FISH used a single fluorophore, TRITC (tetra-methyl rhodamine isothiocyanate), to label RNA [52]. Likewise, Singer and Ward utilized a single Rhodamine conjugated to an antibody in the first iteration of RNA FISH [54]. Later studies employed multiple probes with differing fluorophores to interrogate the relationships and localizations of multiple target DNA sequences simultaneously. This approach resulted in the development of multiplex-FISH (M-FISH), a technique in which multiple targets are simultaneously tagged with up to 24 different fluorophores, enabling clinicians to screen karyotypes for deleterious genetic mutations [56,57]. This system was eventually adapted to RNA with the development of sequential-FISH (seqFISH).

One of the significant problems with mRNA FISH remains the photostability of the fluorophores used, particularly when attempting super-resolution, as the photon flux required in super-resolution environments results in the rapid deterioration of fluorophores during excitation [28,58]. One response to this problem has been to move away from organic fluorophores. An intriguing recent advance in this arena has been the development of quantum dots, which exhibit brightness equal to organic dyes with significantly improved photostability. In fact, fluorescence intensity was observed to be undiminished over a 12-min period of continuous excitation, indicating that this new fluorescent labeling strategy could provide a means to a more durable, robust fluorophore in RNA FISH, enabling a better 3D localization of mRNA localizations in the future.

The second approach to improving upon ISH methodology, the addition of multiple fluorophores to a single probe, was first utilized in RNA FISH by attaching multiple CY3 fluorophores to a DNA probe. This approach resulted in significantly higher fluorescence and enabled researchers to identify discrete sequences of mRNA enabling the quantification of mRNA for the first time via mRNA FISH. Although this advance is often considered the first incarnation of single-molecule FISH (smFISH) [59], this approach faced significant technical challenges, as the close proximity of the CY3 fluorophores resulted in self-quenching [55,60]. This resulted in differential fluorescence intensity of individual probes, making quantification via this methodology unreliable. Although this particular iteration was unable to resolve single particles, the idea of multiple fluorophores generating an additive fluorescent intensity was a good one and eventually resulted in the development of smFISH in 2008 [58].

**Figure 1 cells-11-03079-f001:**
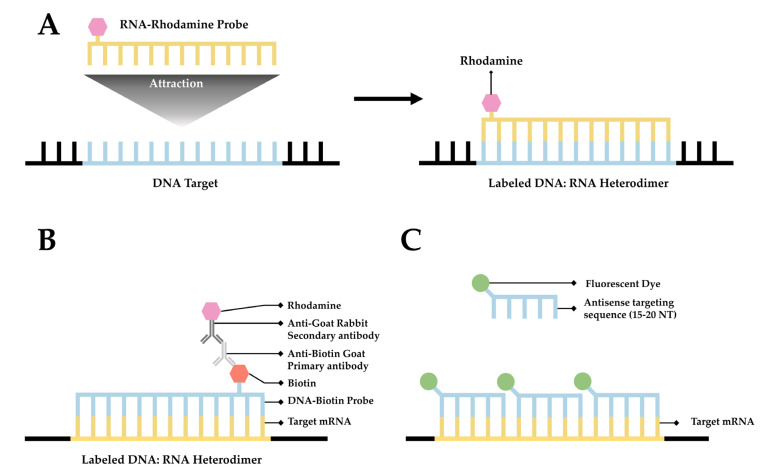
**A simplified diagram depicting fluorescent in situ hybridization (FISH).** (**A**) The first FISH experiment in DNA performed by Bauman and colleagues [52], in which a target DNA sequence (Blue) forms and RNA:DNA heterodimer with an RNA (Green) probe conjugated with Rhodamine (Red). (**B**) The first mRNA FISH experiment performed by Singer & Ward [54], in which a target mRNA sequence (Green) formed a DNA:RNA heterodimer with a complimentary DNA sequence (Blue) conjugated to biotin (purple). A primary anti-Biotin Goat derived primary antibody (Dark Blue) associates with the biotin tag. A secondary anti-Goat rabbit derived antibody (Light Red) conjugated to a Rhodamine (Dark Red) then associates with the primary antibody forming a complete fluorescent label. (**C**) The core principle of single-molecule FISH (smFISH), in which a target mRNA (Green) is targeted with short sequential antisense oligonucleotides (Blue), each 15–20 nucleotides long, that are each conjugated to a fluorescent dye (Dark Green).

### 2.2. smFISH

Single-molecule FISH (smFISH) was developed by Raj and colleagues in 2008 to address limitations in FISH concerning low intensity of signal [58]. The authors tackled this problem by producing a series of 15 to 20 nucleotide long antisense oligonucleotide (ASO) probes, each conjugated to a fluorescent dye (Figure 1C and Figure 2A), along the length of the coding DNA sequence (CDS) and 3′ untranslated region (UTR) regions of intended mRNA transcript. The additive intensity of these multiple probes enabled easier detection of the transcripts such that they could be imaged, localized, and counted (Figure 2A) [58]. Using different combinations of spectrally distinct probes, multiple mRNAs can be imaged simultaneously; however, because of potential spectral overlap, the number of possible combinations is small (See Figure 2A). (The authors demonstrate three distinct mRNAs).

As described by Raj and colleagues, smFISH is quite sensitive because of the number of probes attached to each mRNA. In addition to providing more sensitivity, smFISH provides for a more robust system. The increased number of fluorophores over the single fluorophore found in traditional RNA FISH enables researchers to employ more photo intensive imaging techniques to investigate the 3D localization of the smFISH-labeled mRNA, and is considered by many to be the gold standard of mRNA localization at the organelle level. Further, the technique offers the option of labeling different segments of longer mRNAs with differently colored probes, which can serve to differentiate complete mRNAs from broken or partially degraded transcripts. Overall, smFISH serves as a useful technique in fixed cells for studies that require the detection and relative quantification of two to four mRNAs. This is accomplished by pairing smFISH with quantitative reverse transcription polymerase chain reaction (qRT-PCR). As this method allows a researcher to label the entirety of an mRNA transcript, this technique can be useful in studies where it is important to distinguish full transcripts from damaged or partial ones. This is done by calculating the fluorescent intensity of an individual fluorophore on an individual antisense targeting sequence, and then determining which spots have the appropriate fluorescent intensity commensurate to having a completely labeled and intact mRNA sequence.

While smFISH has many technical strengths and has been invaluable in answering many questions regarding mRNA behavior within the cell, it suffers from many significant shortcomings. This technique is viable only in fixed cells. This presents four significant problems to imaging mRNA. First, static information derived from a fixed cells only provides researchers with a snapshot of their behavior. Second, the fixation of a cell often alters membrane structures, potentially leading investigators to draw potentially erroneous conclusions regarding interactions between mRNA and the nuclear envelope [61]. Third, the assay is technically challenging to perform and requires assay optimization for each target. This is due to the presence of too many ASOs producing untenable background noise. Alternatively, if too few ASOs are present, the data cannot be trusted as the full transcript is unlikely to be labeled. Finally, smFISH relies upon qRT-PCR for relative quantification. This further complicates this already technically demanding method with the addition of another technically demanding technique. Further, it has been well noted in the literature that qRT-PCR may have bias depending on the baseline and efficiency of the reaction [62,63].

### 2.3. seqFISH

Using techniques such as smFISH to resolve and identify mRNAs is limited by their proximity, which is frequently below the optical diffraction limit. This potentially causes spectral overlap and the loss of discrete signal. Whereas temporal separation of detection is a key feature of other single-molecule approaches such as SPEED microscopy [49,64,65,66,67], it requires a low concentration of tagged molecules in the field of detection. For techniques such as SPEED, which is particularly suited to the dynamic environments of live cells, the need to keep the concentration of detected molecules low presents a limitation. In techniques that are more suitable for static environments, such as STORM [33,68] and STED [69], the need for low concentration of probes can be overcome to some degree with the use of multiple colors of fluorophores [70] if fixed cells are used.

Lubeck and Cai in 2012 and Shah and colleagues in 2016 described sequential FISH (seqFISH) as a way to overcome these limitations to result in a technique that resolved transcripts from 32 stress-responsive genes in single *S. cerevisiae* cells by combining spatial and spectral coding using (spatial) order of probes (along a transcript) and combinations of colors [71,72]. Their first approach was combinatorial labeling, in which activator-emitter probes were spaced along the mRNA far enough apart (about 100 nucleotides) to be resolved at the resolution capable with STORM. (The localization resolution of STORM is approximately 20 nm [33].) The authors hybridized probes of varying spectral patterns in particular order to different mRNAs to differentiate them, terming this technique *spatial barcoding* [71] (see Figure 2B). Spectral and spatial barcoding as described by the authors is well suited to identify multiple different mRNAs (the authors claim a limit of 792 genes with an additional probe) to a resolution of approximately 20 nm, providing ample resolution to colocalize mRNA with many subcellular structures. Although this labeling approach is useful in varying situations, it relies on the temporal aspect of STORM to reach its full potential, which can represent a financial obstacle.

**Figure 2 cells-11-03079-f002:**
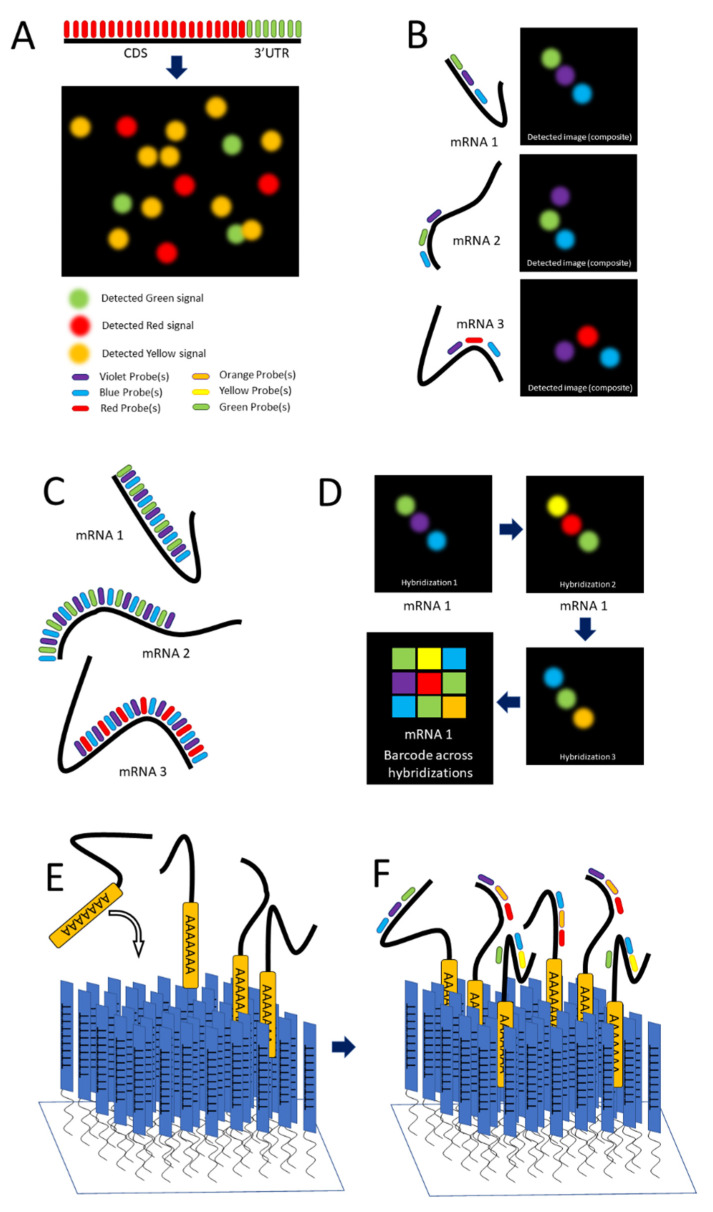
**smFISH and seqFISH.** (**A**) smFISH: Top, multiple probes of the same color designed to hybridize along the length of both the CDS and 3′UTR of the intended transcript; bottom, mRNAs imaged after hybridization, with yellow spots showing transcripts containing both the CDS and 3′UTR [after (Raj, 2008)]. (**B**) seqFISH Spatial barcoding, in which probes are designed to hybridize ~100 nt apart to facilitate resolution of unique combinations (after [71]). (**C**) Spectral barcoding, in which a color code of probes hybridizes repeatedly along the length of a transcript in order to increase its detectability and identifiability. (**D)** Repeated cycles of hybridization, imaging, and removal of probes results in a temporal barcode that increases the number of unique barcodes possible and aids resolution via the temporal dimension (Panels C,D after [73]). **©** seqFISH in vitro: An oligo(dT) surface is created and mRNAs hybridize to it via their poly-A tail, spreading out to a resolvable distance. (**F**) Probes are hybridized to the adhered transcripts.

Lubeck and Cai further refined their approach by labeling mRNAs with repeating probes of a single-color combination (per mRNA), ignoring spatial order. Because transcripts are labeled multiple times throughout their length, partially degraded and nonlinear mRNAs as well as those entangled with proteins and other macromolecules can be detected. These features ease the resolution requirement for confident detection. The authors term this approach *spectral barcoding* [71] (see Figure 2C).

Lubeck and Cai develop seqFish even further by subjecting fixed cells to repeated rounds of hybridization, imaging, and then stripping of the probes by DNAse I, and photobleaching of the remaining signal [73]. In each hybridization, a different colored probe is used. The images are collated in temporal order to create a barcode that identifies unique mRNAs. The combination of colors across hybridizations (essentially across time) replaces spatial and spectral barcoding, removing limitations such that in theory the number of identifiable unique mRNAs is unlimited. By combining sequences of colors with individual transcripts, the number of unique barcodable mRNAs increases exponentially, scaling as F^N^, F being the number of fluorophores and N the number of rounds of hybridization (see Figure 2D). The authors cite four fluorophores and eight rounds of hybridization being sufficient to cover the entire transcriptome [73].

Eng and colleagues further develop seqFISH by pivoting from in situ applications into quantification of mRNAs extracted from cells and tissues, calling this new in vitro application *RNA SPOTs* [74]. The authors used the RNA SPOTs technique to compare the differential expression of genes between mouse fibroblasts and mouse embryonic stem cells (mESCs), finding extracellular matrix and collagen maintenance genes highly expressed in the fibroblasts but not the mESCs and pluripotency factors such as Rex1, Esrrb, and Sox2 highly expressed in the mESCs but not the fibroblasts [74].

Extracted mRNA presents the problem of optical crowding; this obstacle is overcome by the authors via the use of an oligo(dT)-coated coverslip. Oligo(dT) consists of short, single-stranded sequences of deoxythymine. This surface provides a 2D space for mRNAs to adhere and creates some distance between individual transcripts so they can be identified more easily. Oligo(dT) ligands covalently bind to a solid support (in this case the treated coverslip) and hybridize to the mRNA via the poly-adenylated tail. Then, seqFISH is performed as described previously [74] see Figure 2E,F).

RNA SPOTs shows its strength in the sheer number of mRNA species it can identify and differentiate, achieving this by extracting them from the cell and spreading them out on a surface suitable to examination via SLM. Thus, this strategy is well suited to studies in which the identification of a large number of mRNAs is a must (such as in comparing the transcriptomes of two different cell types), but in which subcellular localization or colocalization information is not needed and thus extracted tissue can be used in lieu of fixed cells. However, like other methods discussed here, neither the dynamics of the live cell environment nor the spatial or temporal relationships of mRNA expression in the cell can be studied with this technique.

### 2.4. MERFISH

It is currently desirable to obtain information regarding the transcriptome localization during various cellular states. To this end, multiplexed error-robust FISH (MERFISH) was developed by Chen and colleagues in 2015 [75]. MERFISH relies upon similar methodologies as seqFISH and smFISH, where multiple antisense oligomers are utilized similar to smFISH. The difference between smFISH and MERFISH becomes apparent as this technique adds each fluorescently tagged antisense oligormer in a sequential manner, thereby uniquely labeling RNA in a manner similar to seqFISH. Where seqFISH creates a color barcode, MERFISH relies upon a 16-bit coding approach.

In computer science, a bit is a binary unit that can be read as a 1 or a 0. Similarly, a qualitative off or on designation can also be applied in fluorescent imaging. With this rationale, Chen and colleagues use sequential FISH probes to generate a code word that is unique to each target mRNA, permitting up to 140 code words using the 16-bit MHD4 code or 1001 possible code words using the MHD2 code [75,76]. A single probe is introduced and then imaged, the computer then labels each fluorescent spot with a 1, the computer then records the position and assigns a 1 or 0 designation if the probe fluoresces or not. The entire cell is then photobleached, and the second probe is introduced. This process proceeds until all probes are utilized.

This technique provides researchers with a robust, error-resistant methodology for quantifying and localizing specific RNA of interest within the cell. Similar to smFISH, this method can only be utilized in fixed cells as the immobility of the RNA is critical to the success of this method. This makes this an ideal technique for capturing whole transcriptome information at key points in the cell life cycle, but not well-suited for capturing details regarding RNA dynamics. Further, producing so many smFISH probes is costly and untenable; to address this, the authors have adapted an existing Oligopaint approach [77,78]. This approach also requires a degree of coding competence to be able to both generate the ‘codebook’ of target RNA sequences and automate the localization and association of the individual sequences with their designated code word. Lastly, this approach may be hampered by possible photodamage. Theoretically, this technique can be expanded to cover the entirety of the transcriptome; however, the photobleaching required between each hybridization could potentially damage the target cell after multiple sequential rounds of hybridization and imaging.

### 2.5. Single-Molecule Localization Microscopy (SMLM)

SMLM is a broad discipline that covers a wide array of methods and techniques that have variously been utilized to image RNA. All of these techniques fundamentally rely upon the principles of spatial localization, meaning that sufficient temporal or physical space is required between individual fluorophores to permit the mathematical fitting of the point-spread function (PSF) without overlapping, thereby determining the 2D localization as well as the localization precision [43,46,49]. The separation of individual PSFs can be accomplished utilizing specialized fluorophores such as photoswitching or photoactivated fluorophores for STORM and PALM, respectively. Alternatively, this separation can be achieved by physically controlling the concentrations of the fluorophore to ensure sufficient separation [43].

In addition to generating pointillistic images, SMLM techniques can be utilized to localize and track single particles. This can, in principle, be accomplished using any appropriate fluorophore or technique that enables sufficient temporal and spatial separation permitting the fitting of single particle PSFs at the single-molecule level [79]. Our lab has previously explored these techniques as they relate to tracking single mRNA particles [67,80].

In addition to 2D localization information, some SMLM methodologies are capable of capturing 3D localization information of single particles. This is accomplished in a variety of ways, the most prominent being PSF engineering methodologies. The PSF of a standard fluorescence microscope changes relatively slowly as the Z-dimension changes above or below the focal plane. This can be altered by the addition of a variety of microscope modifications to generate an engineered PSF that will change in relation to the location of the fluorophore to the focal plane [46]. These engineered PSFs are capable of providing Z-dimensional information within a range of 0.8 to 20 μm depending upon the specific PSF engineering methodology employed and includes the following approaches: Astigmatism [81], Phase Ramp [82], Double Helix [83], Accelerating Beam [84], Corkscrew [85], and Tetrapod [86]. In contrast to PSF engineering, SPEED microscopy derives virtual 3D information from the 2D data captured within a rotationally symmetric object, such as the nuclear pore complex (NPC). This is accomplished by running the 2D information through an algorithm that builds a probability density matrix based upon localization within the symmetrical object [49,87]. This technique has been utilized to evaluate the transport of mRNA through the NPC in live cells with a temporal resolution up to 0.4 ms and a localization precision of ~10 nm [48,49,88,89]. For the purposes of this manuscript, it suffices to say that each of these techniques can be utilized to image RNA provided that an appropriate RNA labeling strategy is employed.

## 3. RNA Labeling Strategies

### 3.1. MCP-MS2 Loop System

The MCP-MS2 loop system is the current gold standard for labeling mRNA in living cells and has been utilized for numerous studies in a variety of model systems over the past few decades [41,48,90,91,92,93,94,95,96,97]. This technique is derived from the virus *Emesvirus zinderi* also known as Bacteriophage MS2 (MS2), a virus that stands out for being the first genome ever fully sequenced [98,99]. MS2 is a single positive-strand genomic RNA virus that infects Gram-negative bacteria with a retractile pilus and contains a genome of ~4 kb that encodes for four proteins: the maturation protein, the coat protein, the replicase, and the lysis protein [100,101]. Following entry via pilus, the virion goes through uncoating, exposing its genome. The MS2 genome is then cleaved, translated, and replicated. Much of the MS2 genome produces stem-loop structures following transcription that prevent translation. During assembly, the coat MS2 coat protein (MCP) recognizes these stem-loop RNA structures, binds to them, and facilitates encapsidation [102].

This observed process has provided researchers with a method to label RNA in living organisms. As stem-loop structures do not exist in mammalian cells, MCP does not interact with macromolecules endogenous to mammalian cells. This affords researchers the ability to develop a bipartite labeling methodology, where two plasmids would be introduced to a cell of interest. First, a plasmid containing a mammalian promoter, the sequence encoding for the RNA of interest, and the MS2 loop sequences in the 3′ UTR. Second, a plasmid containing a mammalian promoter, nuclear localization signals (NLS), and the sequence for MCP conjugated with a fluorophore on the 3′ end (Figure 3A). When both plasmids are present and transcribed/translated, an RNA sequence containing MS2 loops at the 3′ end is present, as well as a nuclear-localized MCP-fluorophore. The target sequence and the MCP-Fluorophore then bind with high affinity, creating an MCP-MS2 array that can be localized using light microscopy (Figure 3B). It is important to note that the addition of the NLS to the MCP-Fluorophore enables researchers to ensure that both plasmids are present in the cell, as cells in which fluorescence is only present within the nucleus were lacking the target RNA sequence. This is due to the fact that MCP-Fluorophore binding to mRNA will be exported to the cytoplasm, thereby causing fluorescence, however dim, to be present in the cytoplasm as well as the nucleus (Figure 3C,D).

Early versions of this approach suffered from significant signal-to-noise complications, as unbound fluorescently tagged MCP proteins are ubiquitous within the nucleus. If the chimeric RNA sequence contains a single MS2 sequence, it is impossible to differentiate between MCP associated with RNA and unbound MCP. This problem was addressed by Bertrand and colleagues by adding 24× MS2 loop sequences to the 3′ UTR of their target sequence [90]. By adding 24× MS2 loops to their chimeric RNA, each target has multiple regions for MCP to bind. This enables researchers to differentiate between bound and unbound MCP using relative fluorescent intensity.

Using this system, Mor and colleagues were able to collect some of the first dynamic information regarding nuclear export at the single-mRNP level [103]. Using this method, the export behavior of mRNPs was further interrogated by authors at the single-NPC and single-mRNP levels [41,48,49]. Further, whereas the chimeric mRNA sequence contains 24× MS2 loops, it was shown that MCP does not bind to every available site. In fact, it was observed that per chimeric mRNA sequence, an average of 8 MCP-fluorophores will bind to the 24× available MS2 loops [41]. This number of fluorophores generates sufficient additive fluorescence to facilitate SMLM, but is not sufficient to easily separate single mRNPs from background noise, instead requiring significant hands-on data analysis to differentiate between background MCP and bound MCP [49].

In summation, the MCP-MS loop system provides several distinct advantages detailed here, most notably being the ease of use in live cells. However, there are also many distinct disadvantages to this system. Chief among the disadvantages is the high background noise present in the system, as unbound MCP-fluorophore conjugates aggregate in the nucleus. Next, the MS2 loop, as it is not present in mammalian cells, confounds cellular machinery. Chimeric mRNA transcripts are untranslatable, and the tight bonding between the MCP-MS2 loop impairs accessibility of mRNA decay enzymes to the MS2 array, leading to slow degradation in *S. cerevisiae* [104,105,106,107]. This deficiency presents significant difficulty for researchers interrogating the full life cycle and intracellular localization of mRNA. The issue has been addressed by developing new versions of the MS2 loop system, specifically by adjusting the linker space between the individual stem-loops. This adjustment is hypothesized to create more space to allow endogenous proteins to interact with the chimeric sequence, thereby reducing or potentially ameliorating this impairment [108,109].

### 3.2. Antibody Labeling

Several steps within the life cycle of an mRNA molecule may inhibit or prematurely end its progression to the ribosome. One of these inhibitory instances, occurring during transcription, is the formation of an RNA:DNA hybrid, R-Loop, from single-stranded RNA hybridization with the complementary DNA sequence. Numerous factors may cause RNA:DNA hybrid formation, relaxed upstream supercoiling, defective proteins related to stalled transcription and RNA:DNA hybrid resolution, G-rich mRNA regions, and non-template DNA strand nicks and secondary structures [110]. Typically, RNA:DNA hybrids are resolved by endogenous RNase H. Although RNA:DNA hybrid formation is not limited to mRNA, they have a significant role in gene expression through CpG island promoters, which encode housekeeping genes and terminator regions [111].

#### 3.2.1. RNA:DNA Hybrids and S9.6 Antibody Labeling 

The complexity of the transcriptome limits the techniques used in microscopy-based assays of mRNA localization and quantification. Typically, localization of a specific mRNA occurs through in situ hybridization with either a fluorescently labeled complementary sequence or a tag recognized by a fluorescently labeled antibody [24]. However, the dynamic nature of RNA:DNA hybrids as a response to genomic instability limits probe targeting by specific RNA sequences. So, a more general method was developed using direct recognition by fluorescently labeled antibodies. Initially reported by Boguslawski et al., the S9.6 mouse monoclonal antibody (mAb) has been a benchmark probe for decades in RNA:DNA hybrid research [112]. Because of the non-specific nature of the S9.6 mAb, it has been used in various techniques to isolate, identify, and image RNA:DNA hybrids [113,114,115]. Conversely, a significant disadvantage to having numerous binding targets is a lack of specificity, resulting in off-target binding, specifically between double-stranded RNA (dsRNA) [116,117]. To neutralize the off-target binding by the S9.6 mAb, several exogenous RNase enzymes are added to degrade endogenous dsRNA and ssRNA in a fixed and permeabilized cell environment. However, the RNase enzymes may also degrade RNA:DNA hybrids [115].

Additionally, S9.6 antibody off-target binding may result from the labeling method used to fluorescent label S9.6 mAb. Commonly, to expose and label cysteines with a fluorophore by maleimide derivatization, the disulfide bonds within the hinge-region are reduced with a mild reducing agent. From this reduction, the antibody is halved and may exhibit a loss of affinity in the antigen binding regions [117,118,119]. Although S9.6 mAb may be labeled without fragmentation, the reproducibility of any experiment may be impacted by the broad spectrum of labeling options. An advantage of antibody fragmentation is reducing the approximately 150 kDa S9.6 mAb to 25 kDa [117], significantly improving its accessibility to centrally located binding targets. 

#### 3.2.2. New Probes for RNA:DNA Hybrid Labeling 

Utilizing the same protein for binding and degrading hybridized RNA in RNA:DNA hybrids, RNase H, researchers have successfully inactivated the catalytic activity of the hybrid binding domain (HBD) [120]. Additionally, HBD binds RNA:DNA hybrids 25–30-fold greater than dsRNA [120,121], improving on the near-equal affinity for RNA:DNA hybrids and dsRNA of S9.6 mAb [117]. Interestingly, because the HBD is an isolated region of endogenous RNase H, Bhatia et al. successfully transiently transfected EGFP-labeled HBD into HeLa cells [122]. Revealing the indirect association of mRNP and TREX-2 to play a role in RNA:DNA hybrid formation prevention via tumor suppressor BRCA2. Because of the accumulation of RNA:DNA hybrids in BRCA2-depleted cells, the authors concluded RNA-mediated genomic instability was a driving factor in cancer-related cellular stress. Furthermore, by directly labeling HBD with a fluorescent protein, they removed any potential issues with labeling inconsistencies and loss of function from fragmentation. Furthermore, with an approximate molecular weight of 35 kDa, the EGFP-labeled HBD may localize centrally located RNA:DNA hybrids.

### 3.3. Multiply-Labeled Tetravalent RNA Imaging Probes (MTRIPS)

Imaging endogenous mRNA is preferred as it provides a more accurate picture of biological processes unaffected by plasmid overexpression or other experimental artifacts arising from transfection. Further, endogenous mRNA also avoids restriction to cell types that can be efficiently transfected. Studying mRNA expression in live cells can provide dynamic information about how mRNA expression changes in response to varying conditions or over time. 

Santangelo and colleagues developed multiply labeled tetravalent RNA imaging probes (MTRIPs), a method of labeling native mRNA transcripts with multiple fluorophores [123]. Synthetic oligos (2′ O-methyl RNA-DNA chimera nucleic acid ligands) were labeled with multiple fluorophores, bound to streptavidin, and delivered into the live cell via reversible permeabilization with streptolysin O. After entry into the cytosol, multiple ligands bind to target mRNA transcripts (see Figure 4). Unbound probes and mRNAs bound to few probes can be eliminated by measuring differential intensity; points of light showing the intensity level expected for one probe are disregarded; mRNAs bound to multiple probes show a higher signal-to-noise ratio. The authors leveraged the technique they developed to image RNA at the single-molecule level, showing colocalization of RNA with RNA-binding proteins in live human epithelial cancer cells and primary chicken fibroblasts [123]. Later innovation resulted in the development of proximity ligation assays utilizing MTRIPS [124], thereby providing more versatility to this labeling strategy.

### 3.4. CRISPR-Based Labeling Strategies

Using endogenous tagging of mRNA such as the previously mentioned MS2 loop systems would be further improved by the ability to shorten the endogenous tag insertion steps and flexibility in insertion sites. Clustered Regularly Interspaced Short Palindromic Repeats (CRISPR) is a hot topic field in genomic manipulation, first discovered in 1987 [125] and pushed into development as a tool for genetic modifications concurrently by Charpentier [126] and Zhang [127]. Using a sequence-specific targeting Single guide RNA (sgRNA), we can precisely target excision sites to insert exogenous tags into th’ 3’ UTR of mRNA sequences [128,129]. Many online tools have been developed to help find appropriate sgRNAs and protospacer sequences to target virtually any gene: “Zhang Lab.” Available online: http://crispr.mit.edu (accessed on 8 August 2022), “Benchling.” Available online: http://benchling.com (accessed on 8 August 2022). These tools are most effectively used in conjunction with Zhang’s protocol for assembling gRNA plasmids [130]. 

The CRISPR/Cas9 system is small enough to fit onto small format delivery mechanisms such as electroporation of mRNA nanoparticles [131,132] and adeno-associated virus (AAV) [133]. The specificity and ease of tag insertions allow for novel multiplexing of single-mRNA labeling of several gene targets at once [134] and flexible modifications [135]. Single mRNA labeling of high abundance mRNA is becoming more trivial with CRISPR technology; Han has developed a novel integration with SunTag [136] for imaging endogenous low-abundance mRNAs [137]. Whereas Cas9 has been primarily utilized for its DNA binding affinity, it has been seen in conjunction with single-site mismatches in respective protospacer adjacent motif (PAM) sequences to potentially target mRNA without affecting protein expression levels [135,138] (Figure 5). An excellent overview that touches on the utility of dCas9-mRNA binding was recently published [139].

Several classes of Cas have been discovered that have an affinity to RNA binding, specifically Cas13 [140,141,142,143]. Cas13 has been utilized in recent years for the direct labeling of mRNA [144], as Yang et al. used this approach to visualize mRNA directly in vivo [145,146]. The dCas13 technique allows real-time RNA imaging without the need for genetic modification. An optimized dCas13 method yields equivalent RNA-labeling efficiency and is user-friendly, free of the need for genetic modification, and superior to the aptamer-based MS2-MCP technique. Chen expanded to multiple gene mRNA using dPspCas13b and dPguCas13b and showed the ability to multiplex in conjunction with dCas9 [147]. This shows alternative Cas systems that are RNA-specific can provide a valuable tool in excising the need for genomic modifications to perform our previously mentioned technologies for single-molecule localization and tracking of mRNA (Figure 5). 

### 3.5. RNA Molecular Beacons

An RNA targeted molecular beacon functions along a similar premise as all ISH methodologies but is especially similar to smFISH. Specifically, a short, 15- to 20-nt, fluorescently tagged DNA oligo binds to a complimentary transcript of interest within the cell, thereby enabling researchers to visualize and localize that transcript. The primary difference is that this technology attempts to resolve the background noise problem inherent to so many of the mRNA labeling techniques discussed in this manuscript. Researchers have accomplished this by adding a fluorophore to one end of the sequence and a quencher to the other [148] (Figure 6A). A quencher is a compound that, when sufficiently close to a fluorophore, will absorb the energy released by the fluorophore and dissipate it as heat [148,149]. The quencher is utilized to great effect by designing the beacon as a stem-loop, flanking the 15- to 20-nt antisense target sequence. When the beacon is not bound to its target, the beacon forms a hairpin, bringing the fluorophore and quencher into close proximity (Figure 6A), thereby generating a probe that will only fluoresce when it is bound to the target (Figure 6B) [150].

This probe can be used in vivo or in situ, making it a standout from other ISH techniques. The probe can be introduced into live cells through a variety of methods, including electroporation, gene guns, microinjection, and digitonin membrane degradation [149,151,152]. This technique is very useful as it enables researchers to track mRNA in live cells, thereby obtaining dynamic information regarding the single molecule localization of a target mRNA sequence. Further, the quencher-fluorophore pair, when in its stem-loop form produces extremely low background noise. This is beneficial to researchers as it provides a high degree of confidence in subcellular localizations derived using this technology. In addition to the more traditional usage of a fluorescent tag for mRNA imaging, researchers have recently used this technology to develop rapid assays to detect SARS-CoV-2 mRNA in patient samples [153].

### 3.6. RNA Aptamer

An aptamer is a single-stranded length of DNA or RNA that forms a secondary structure that selectively binds to a specific target [154]. As there are no known fluorescent RNAs, this is an attractive feature for imaging RNA. Conditional fluorophore aptamers, also known as fluorescent turn-on aptamers, have been developed to enable the imaging of RNA. These aptamers form a specific secondary structure which will then selectively bind to a specific dye that exhibits minimal fluorescence until it binds with its cognate aptamer. This is made possible as a result of the following three principles, twisted intramolecular charge transfer, excited state proton transfer, and unquenching of fluorophore-quencher conjugates [155]. Each of these results in a physical change to the dye enabling it to emit fluorescence upon excitation when associated with the aptamer. 

This system has been utilized in a variety of studies to evaluate the localization and behavior of telomerase-associated RNA in both mammalian cells and *Saccharomyces cerevisiae* [156,157,158,159]; specifically, TLC1 [160], the RNA scaffold for the telomerase holoenzyme, and telomeric repeat-containing RNA (TERRA), a DNA:RNA heteroduplex that actively participates in the telomere maintenance and chromosome end protection [161]. Providing valuable information regarding the localization and dynamics of these RNAs within both dividing and senescent cells.

In summation, this system provides an attractive option for labeling RNA in fluorescent microscopy. In many ways, this approach can be combined with CRISPR-based strategies, as a degree of genomic engineering is required to ensure that the sequence of interest is labeled appropriately. This system also has a few significant shortcomings. Specifically, the early generation of spinach and broccoli aptamers suffered from poor folding in live cells, low quantum yields, and rapid photobleaching [156]. However, these challenges were partially overcome in later iterations with the Mango and SiRA lines of aptamers [156,162,163].

## 4. Perspective

There are a wide variety of techniques for labeling mRNA for light microscopy, both for the cellular level and single-molecule level. Each of which have their own strengths and weaknesses (Table 1). These techniques provide invaluable information for the researcher, particularly as it relates to relative localizations, transcription quantifications, and even the visualization of the transcriptome. However, there remain many avenues for improvement of these labeling techniques, as there are currently few techniques for accurately imaging the transcription, processing, packaging, export, and translation of mRNA. Many of these techniques are only possible in permeabilized or fixed cells, where experimental conditions are optimized and not indicative of real-world scenarios. Further, fixed cells especially are limited by an inability to capture dynamic information. Perhaps the field that requires the most improvement is labeling techniques within living cells.

Living-cell mRNA labeling presents a particularly difficult challenge when compared with protein labeling. The advent of genetic engineering made it possible not only to transfect cells with plasmids containing fluorescent proteins, but to modify endogenous proteins with a fluorescent protein. Because mRNA, by its very nature, is untranslated, there remains a significant challenge in imaging endogenous RNA to answer questions regarding native RNA behaviors. Some techniques have significantly contributed to resolving these questions; chief among them is the MCP-MS2 loop system. However, this system remains hampered by the two-part system requiring transfection with the MCP plasmid. This introduces two levels of uncertainty. First, MCP that is unbound is present and free floating through the nucleus. This free-floating property creates an unacceptably high background fluorescence, making it difficult to isolate nuclear neighborhoods, track single molecules, or observe behaviors related to splicing and packaging for export. However, this problem is not insurmountable as a super-folder GFP system has been utilized as a potential solution to this issue [164]. Second, the act of transfection, either through electroporation or lipofection, inherently changes the state of the cell. Together, these techniques potentially introduce novel behaviors to macromolecules of interest that may lead to erroneous conclusions. Molecular probes have provided a rather unique way to bypass the first challenge, that of background noise, but remains still hampered by the second. The development of new labeling or imaging techniques is necessary to more accurately evaluate the behaviors of endogenous mRNA. 

An intriguing development in the field of RNA study is the development of proximity labeling techniques. These techniques rely upon enzyme-catalyzed in vivo reactions that occur between a labeling enzyme and a protein of interest [165]. A recent advance employs an engineered ascorbate peroxidase enzyme (APEX2) that converts a cell permeable biotin-tyramide substrate into a highly reactive free radical that labels aromatic amino acids in proteins within ~25 nm [165]. In light of the fact that nucleotides are amenable to free-radical-based chemistry [165,166] a system utilizing APEX2 to label RNA was developed, called APEX-seq. This system is relatively new and its full capabilities have yet to be explored; however, proximity based labeling of RNA provides an intriguing possibility for future research. Researchers could potentially utilize this system to track and quantify the relative number of specific mRNAs passing in close proximity to the engineered enzyme. For example, utilizing this system in the nuclear basket of the NPC could provide intriguing information regarding the docking and export behaviors of mRNA.

Of particular interest in recent years is the idea of an RNA-involved phase-separation. This can be seen during transcription where it is hypothesized that RBPs interact and synergize the phase separation of polymerase condensates to promote transcription [167]. To investigate this possibility, the development of methods to fluorescently tag and utilize SMLM techniques is required. This will enable researchers to observe these RNA-involved phase-separations interactions in real-time. In particular, a fluorophore is required that is only active during specific physiological conditions found during the phase separation hypothesized to occur at the transcription site. Likewise, the phase-phase liquid separation found within the nuclear pore complex (NPC) also presents a very intriguing area for mRNA imaging. A recent study from Li and colleagues has examined the interaction of mRNA with discrete proteins within the nuclear basket of the NPC [41]. However, current methods do not permit an evaluation of the interaction between mRNA and the phase-phase separation of the central channel of the NPC. The capabilities exist to further evaluate mRNA interactions between different proteins within the NPC, but the development of a fluorophore or labeling strategy to visualize the impact of phase-phase separation on mRNA as it moves through the central channel of the NPC could provide significant insight into the behaviors of RNA during nuclear export.

## Figures and Tables

**Figure 3 cells-11-03079-f003:**
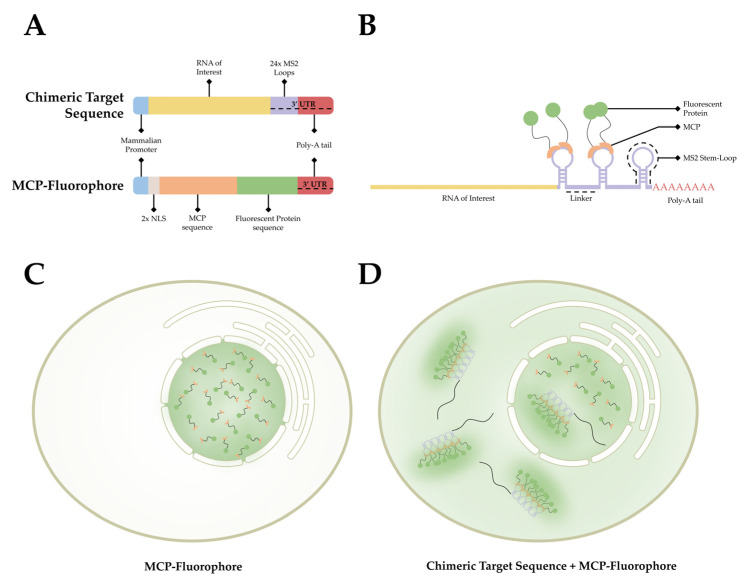
**The MCP-MS2 loop system.** (**A**) A depiction of the gene cassettes present in the two plasmids, the chimeric target sequence and the MCP-fluorophore, utilized in this system. (**B**) A simplified diagram of the association between the chimeric target sequence and the MCP-Fluorophore post transcription/translation. (**C**) A depiction of the fluorescent pattern observed in cells that have only the MCP-Fluorophore plasmid. (**D**) A depiction of the fluorescent pattern observed in cells that contain both the MCP-Fluorophore and Chimeric Target Sequence plasmids.

**Figure 4 cells-11-03079-f004:**
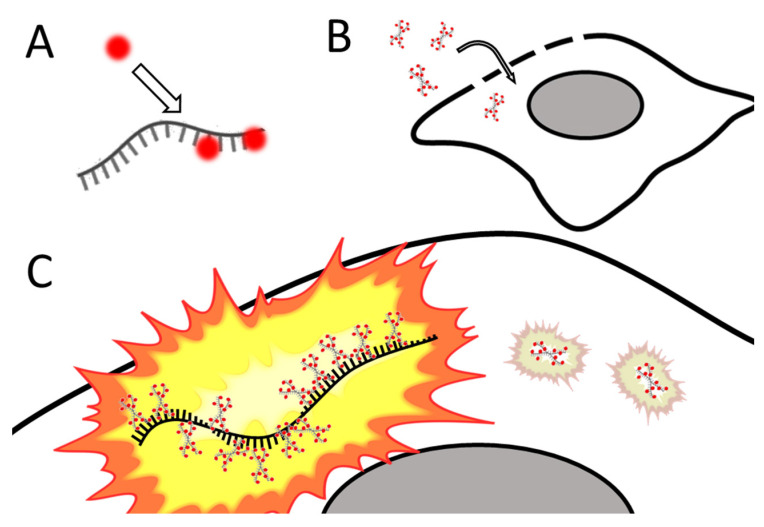
**Simplified schematic of MTRIPs.** (**A**) Fluorophores are bound to synthetic oligomers (ligands). (**B**) Tagged ligands are introduced into the live cell via temporary permeabilization. (**C**) Ligands bind to target mRNA (not o scale); multiply bound mRNAs are distinguished from unbound probes by intensity. After [123].

**Figure 5 cells-11-03079-f005:**
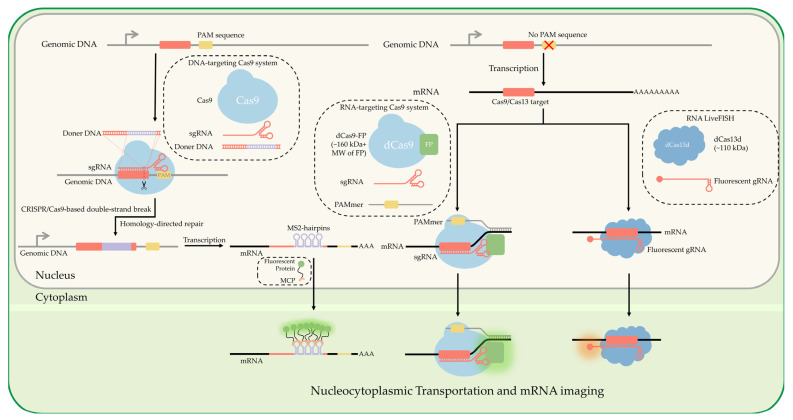
**Nucleocytoplasmic transportation and mRNA imaging utilizing CRISPR.** CRISPR/Cas9 is guided to DNA via sgRNA and neighboring PAM. Nuclease activity of Cas9 makes a double-stranded break, allowing for insertion of targeting aptamers such as MS2 hairpin loops, and for a fluorescent protein fused to MCP to label the modified mRNA. A nuclease deficient Cas (dCas) protein fused with a fluorescent protein can be used to bind to mRNA of interest directly instead, without genetic modification. By providing a double-stranded PAMmer with dCas9, dCas9 can be made to bind with mRNA instead of DNA. Alternatively, dCas13 natively binds to mRNA without the need for PAMmer sequence.

**Figure 6 cells-11-03079-f006:**
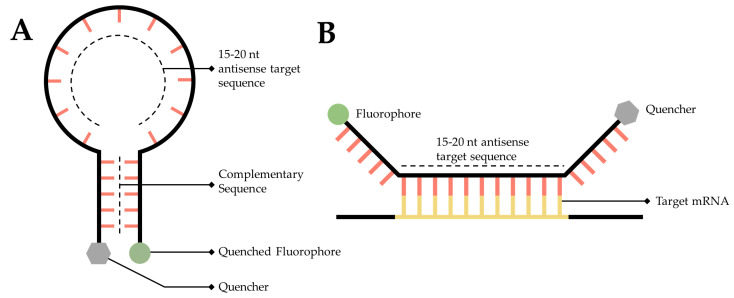
**Simplified diagram of RNA molecular beacons.** (**A**) A 15- to 20-nucleotide target sequence flanked by palindromic repeats causes the probe to form a stem-loop, bringing the quencher (Grey) and a fluorophore into close proximity causing the fluorophore to quench (Dark Grey). (**B**) When in close proximity to the target transcript, the target sequence will hybridize with the target mRNA, causing the stem-loop to open, moving the quencher and fluorophore away from one another, thereby facilitating fluorescence.

**Table 1 cells-11-03079-t001:** **A Non-exhaustive Summary of the Advantages and Disadvantages of techniques and labeling strategies.** This table provides a brief summation of key advantages and disadvantages of key techniques and labeling strategies discussed in this manuscript.

*Methodology*	Single-Molecule Precision	Live-Cell Imaging	Signal to Noise Ratio	Challenging Technique	Quantitative Measurement	Other Advantages	Other Limitations
*FISH*	**No**	**No**	**Poor** Potential off-target interactions and high background noise	**No** Established technique that is well characterized and optimized	**No** Provides only an ensemble average of detected fluorescence	Able to multiplex and label up to 24 different targets with different fluorophores using M-FISH.	Severely limited by photostability of the fluorophores used
*smFISH*	**Yes** When combined with an appropriate technique	**No**	**Good** Multiple probes binding a single target provides higher signal-to-noise ratio	**Yes** Technically challenging	**Yes** Combing this technique with qRT-PCR provides quantification of RNA.	Provides spatial localization at the cellular level.	Cannot localize within subcellular compartments.
*seqFISH*	**Yes**	**No**	**N/A** Requires optimization to visualize SPOTS.	**Yes** Time consuming	**Yes** Capable of relative quantification of RNA.	Can differentiate a theoretically unlimited number of mRNA species	Repeated rounds of imaging may cause photodamage.
*merFISH*	**Yes**	**No**	**N/A** Requires optimization to localize probes.	**Yes** Requires a degree of coding knowledge as well as optimization.	**Yes** Capable of relative quantification	Capable of imaging the full transcriptome using a 16-bit coding approach.	Multiple rounds of photobleaching may cause photodamage.
*RNA Aptamer*	**Yes** When combined with an appropriate technique	**Yes**	**Good** Fluorescent Dyes only emit fluorescence when bound to Aptamer.	**No** Widely used and well characterized	**Yes** Provides relative quantification of RNA.	A wide variety of aptamer probes are available that cater to specific requirements.	Different aptamers have different viability. Many suffer from poor folding, poor quantum yield, and rapid photobleaching.
*MCP-MS2 loop System*	**Yes** When combined with an appropriate technique	**Yes**	**Poor** High background Noise	**No** Widely used and well characterized	**No** Provides only an ensemble average of detected fluorescence	Widespread use allows for many readily available plasmids utilizing this system.	Can interfere with mammalian cellular processes.
*MTRIPS*	**Yes** When combined with an appropriate technique	**Yes**	**Good** Multiple probes binding a single target provides higher signal-to-noise ratio	**Yes** Requires significant post-imaging processing.	**Yes** Provides relative quantification of RNA and associated proteins.	Capable of use proximity ligation assays.	Requires membrane permeabilization to introduce the probe to the cell.
*CRISPR based labeling strategies*	**Yes** When combined with an appropriate technique	**Yes**	**Poor** High background Noise	**Yes** Technically challenging.	**No** Provides only an ensemble average of detected fluorescence	Highly robust and versatile system. Can either engineer a tag into RNA or utilize fluorescent gRNA to label RNA.	Can potentially interfere with mammalian cellular processes.
*Molecular Beacons*	**Yes** When combined with an appropriate technique	**Yes**	**Good** Low background noise	**No** Straightforward and streamlined technique	**Yes** Provides relative quantification of RNA.	Highly specific to the target sequence.	May interfere with cellular machinery. Requires membrane permeabilization to introduce into the cell.

## Data Availability

Not applicable.

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
