# Peer review of "Technologies Enabling Single-Molecule Super-Resolution Imaging of mRNA"

_cells, 2022, doi:10.3390/cells11193079_

Round 1

Reviewer 1 Report

The authors provide insights to the critical role of imaging for studying RNA, and discusses a number of RNA imaging technology in much detail. However, I think the review would benefit from a broader perspective. Some technologies are not mentioned. There are also issues with the flow of the content. The sections are not organized in the most logical manner, and I found it hard to follow why certain technologies are grouped together.

Major comments:

1.       I think part of the issue with the organization of the article stems from the ambiguous use of “single-molecule” in various context (throughout the text, e.g. title, section 1.4, 1.5, 1.6, 3.).
As I interpret conventional understanding:

·         Single-molecule localization microscopy (SMLM) is a super-resolution microscopy method based on the imaging of individual fluorophores. It is NOT inherently a single-molecule method in that molecular resolution is not guaranteed. The majority of SMLM performed do not achieve molecular resolution.

·         Conventional single-molecule imaging/ single particle tracking existed before super-resolution microscopy methods were developed. Typically refers to imaging of single fluorophores. Not traditionally referred to as ‘super-resolution’, achieves high precision and makes no claims in regards to resolution. Not equivalent to SMLM (Line 144, 198).

·         Single molecule RNA imaging refers to imaging of single RNA molecules regardless of how many fluorophores are attached. Can be either diffraction limited or super-resolution imaging.

Due to the ambiguity I don’t really know what section 3 “Labeling for single-molecule super-resolution microscopy” is actually referring to. For example, why is seqFISH described in section 2 but smFISH in section 3.

2.       Along similar lines, “bulk RNA imaging” is a non-conventional term and it is unclear what the authors mean exactly.

3.       The authors describes their SPEED microscopy method, and the MFM method in detail but do not mention other single molecule tracking technologies used for RNA imaging. E.g. Double helix or astigmatism PSF engineering.

4.       The authors do not mention MERFISH, RNAscope, fluorophore binding aptamers, etc. I don’t see a clear reason why these technologies should be omitted. If word limit is an issue, I would argue some of the discussion on labelling methods can be more concise, for example, for the MCP-MS2 loop system.

5.       Line 177, Line 657. The authors refers to fixed cells as in situ and live cells as in vivo. The convention is that both fixed or live cellular imaging is in situ. Whereas imaging of live cells in a living organism is in vivo.

Minor comments:

1.       The clarity of the abstract may benefit from some revisions. E.g. “…traditional concerns associated with sub-diffraction-limit imaging..”. What traditional concerns? In this context do you mean diffraction-limited imaging? “Sub-diffraction-limit imaging” was a term used in the first STORM paper, but otherwise uncommonly used. It is also never repeated in the text.

2.       Line 106. Fluorophore brightness is a function of both quantum yield and extinction coefficient.

3.       Line 113, Line 276. Potential PAINT-based method that may be worth mentioning (10.1093/nar/gkaa623)

4.       Line 131. Calling the diffraction limit a ‘physical barrier’ and ‘impenetrable’ may be misleading.

5.       Line 254. Conventional knowledge is that multiply secondary antibodies can bind a single primary antibody, and that multiply fluorophores are conjugated onto a single antibody. Unless explicitly stated otherwise I would assume there is >1 fluorophore per RNA.

6.       Line 291. I would not call the method ‘failed’. Otherwise most methods paper ‘failed’ because additional improvements can be made.

7.       Line 292. Missing reference for smFISH here.

8.       Figure 3A. Fluorescent protein sequence instead of fluorophore. Figure 3B-D Fluorescent protein instead of fluorophore.

9.       Figure 3C-D seems unnecessary. Is the banding in Figure 3D an artifact?

10.   Figure 5. Too small and unreadable.

11.   Line 619. RNA molecular beacon is a type of molecular beacon. Not all molecular beacon are RNA-based.

12.   Line 629. Palindromic sequences are not compulsory.

13.   Figure 6A. Repeat sequences are not compulsory.

14.   Line 671. Split GFP is a possible solution. (10.1261/rna.067835.118)

15.   Line 683. Grammar/spelling issues.

Author Response

Reviewer 1

The authors provide insights to the critical role of imaging for studying RNA, and discusses a number of RNA imaging technology in much detail. However, I think the review would benefit from a broader perspective. Some technologies are not mentioned. There are also issues with the flow of the content. The sections are not organized in the most logical manner, and I found it hard to follow why certain technologies are grouped together.

Thank you for taking the time to review this article. Your comments have been very helpful and were carefully reviewed and used to revise the manuscript.

Major comments:

  1. I think part of the issue with the organization of the article stems from the ambiguous use of “single-molecule” in various context (throughout the text, e.g. title, section 1.4, 1.5, 1.6, 3.).
    As I interpret conventional understanding:
  • Single-molecule localization microscopy (SMLM) is a super-resolution microscopy method based on the imaging of individual fluorophores. It is NOT inherently a single-molecule method in that molecular resolution is not guaranteed. The majority of SMLM performed do not achieve molecular resolution.

Thank you for this comment. We have clarified that not all SMLM methods will achieve molecular resolution.

  • Conventional single-molecule imaging/ single particle tracking existed before super-resolution microscopy methods were developed. Typically refers to imaging of single fluorophores. Not traditionally referred to as ‘super-resolution’, achieves high precision and makes no claims in regards to resolution. Not equivalent to SMLM (Line 144, 198).

Thank you for this observation, we have clarified this in the manuscript.

  • Single molecule RNA imaging refers to imaging of single RNA molecules regardless of how many fluorophores are attached. Can be either diffraction limited or super-resolution imaging.

A sentence has been added to clarify that single-molecule localization is referring to the single RNA macromolecule.

Due to the ambiguity I don’t really know what section 3 “Labeling for single-molecule super-resolution microscopy” is actually referring to. For example, why is seqFISH described in section 2 but smFISH in section 3.

Thank you for this feedback. We have re-ordered the manuscript accordingly. To resolve these issues we have changed the organization from level of resolution to be RNA imaging techniques and RNA labeling strategies.

  1. Along similar lines, “bulk RNA imaging” is a non-conventional term and it is unclear what the authors mean exactly.

Thank you for this feedback. We have changed the term bulk RNA imaging to reflect that we mean cellular level resolution, not single-molecule resolution.

  1. The authors describes their SPEED microscopy method, and the MFM method in detail but do not mention other single molecule tracking technologies used for RNA imaging. E.g. Double helix or astigmatism PSF engineering.

Thank you for this comment. This issue has now been remedied.

  1. The authors do not mention MERFISH, RNAscope, fluorophore binding aptamers, etc. I don’t see a clear reason why these technologies should be omitted. If word limit is an issue, I would argue some of the discussion on labelling methods can be more concise, for example, for the MCP-MS2 loop system.

Thank you for this comment. MERFISH and RNA aptamers have now been added. RNAscope appears to be utilized primarily in tissue samples and is therefore outside the scope of this manuscript.

  1. Line 177, Line 657. The authors refers to fixed cells as in situand live cells as in vivo. The convention is that both fixed or live cellular imaging is in situ. Whereas imaging of live cells in a living organism is in vivo.

Thank you. This has now been corrected.

Minor comments:

  1. The clarity of the abstract may benefit from some revisions. E.g. “…traditional concerns associated with sub-diffraction-limit imaging..”. What traditional concerns? In this context do you mean diffraction-limited imaging? “Sub-diffraction-limit imaging” was a term used in the first STORM paper, but otherwise uncommonly used. It is also never repeated in the text.

Thank you for this comment. The text has been adjusted to say super-resolution instead.

  1. Line 106. Fluorophore brightness is a function of both quantum yield and extinction coefficient.

Thank you. This section has been adjusted to include extinction coefficient as well.

  1. Line 113, Line 276. Potential PAINT-based method that may be worth mentioning (10.1093/nar/gkaa623)

Thank you for this suggestion. We have now cited this paper.

  1. Line 131. Calling the diffraction limit a ‘physical barrier’ and ‘impenetrable’ may be misleading.

Thank you for this comment. The text has been adjusted to reflect this.

  1. Line 254. Conventional knowledge is that multiply secondary antibodies can bind a single primary antibody, and that multiply fluorophores are conjugated onto a single antibody. Unless explicitly stated otherwise I would assume there is >1 fluorophore per RNA.

We are in agreement with you. As this is a review article written for both experts and non-experts, the authors thought it would be a good idea to mention this.

  1. Line 291. I would not call the method ‘failed’. Otherwise most methods paper ‘failed’ because additional improvements can be made.

Thank you for this feedback. That is a very good point. We have adjusted the text to reflect that the specific method was unable to resolve single-molecules and removed the word ‘failed.’

  1. Line 292. Missing reference for smFISH here.

Thank you for catching this. The appropriate citation is now in place.

  1. Figure 3A. Fluorescent protein sequence instead of fluorophore. Figure 3B-D Fluorescent protein instead of fluorophore.

Thank you for this note. The figure has been appropriately updated.

  1. Figure 3C-D seems unnecessary. Is the banding in Figure 3D an artifact?

Figures C and D were added intentionally to highlight for researchers the fluorescent pattern they will observe wwhen both plasmids have been successfully transfected. The banding in the word document is an artifact caused by the format. The original .tif, which will be used for publication, displays a faint green speckling. To better highlight this, the figure has been further revised.

  1. Figure 5. Too small and unreadable.

Fig 5 has been changed to be in landscape and fill the full page. It should now be large enough to be readable.

  1. Line 619. RNA molecular beacon is a type of molecular beacon. Not all molecular beacon are RNA-based.

The Manuscript has been updated to reflect this.

  1. Line 629. Palindromic sequences are not compulsory.

The manuscript now notes that the addition of palindromic sequences is not compulsory.

  1. Figure 6A. Repeat sequences are not compulsory.

The figure now has these labeled as ‘optional palindromic repeats.’

  1. Line 671. Split GFP is a possible solution. (10.1261/rna.067835.118)

Thank you for this comment. The manuscript now states this system is a potential solution for the high background issue in MCP-MS2 loop systems for mRNA.

  1. Line 683. Grammar/spelling issues.

Thank you for this comment. The sentence in question has been split into two parts and rephrased for grammar, spelling, and clarity.

Reviewer 2 Report

In this review, the authors have nicely elaborated on the different methodologies that are used for RNA imaging and characterization/quantification. The authors have touched upon all the cutting-edge technologies including MCP-2-MS2 labeling, CRISPR-based labeling strategies and Molecular beacons that are currently used to image RNA in live cells. The review is well written and should be of interest to readers involved in studying RNA.

 I’ve just a minor suggestion to the authors:

1.    The authors should summarize all the different techniques in a table format highlighting the benefits and the limitations of each technique.

2.    The authors should also comment on the newly developed proximity labeling by APEX-Seq that maps subcellular RNA localization and protein association in mammalian cells.

Author Response

Reviewer 2

In this review, the authors have nicely elaborated on the different methodologies that are used for RNA imaging and characterization/quantification. The authors have touched upon all the cutting-edge technologies including MCP-2-MS2 labeling, CRISPR-based labeling strategies and Molecular beacons that are currently used to image RNA in live cells. The review is well written and should be of interest to readers involved in studying RNA.

Thank you for taking the time to review this manuscript. Your efforts are greatly appreciated.

 I’ve just a minor suggestion to the authors:

  1. The authors should summarize all the different techniques in a table format highlighting the benefits and the limitations of each technique.

A table highlighting the advantages and disadvantages of the key techniques has been added.

  1. The authors should also comment on the newly developed proximity labeling by APEX-Seq that maps subcellular RNA localization and protein association in mammalian cells.

A short commentary on proximity based labeling, specifically APEX-seq, has been added to the perspective section.

Round 2

Reviewer 1 Report

The authors have responded to most of my comments in their letter but I am unable to confirm all changes were actually made in-text. Please include line numbers in your response and highlight all modified text to expedite the revision process.

Notably, the authors have reorganized the sections. It is easier to follow now, although I probably don’t fully agree with the imaging/labelling grouping. The authors have also added sections on MERFISH, RNA aptamer and APEX.

Major comments:

1.       Please do not introduce a new acronym that directly conflict with an existing, widely popular acronym. SMLM is widely accepted to be single-molecule localization microscopy.
The revised text attempts to circumvent my previous concerns with the ambiguous use of SMLM by renaming the acronym to have a boarder meaning. That was insufficient to fix the problem, the ambiguousness is arguably worst. The authors need to clearly distinguish between when they are referring to single-molecule imaging vs single-molecule localization microscopy.

2.       The revised text requires proofreading. For example, ‘dimension’ has only 1 ‘m’.

3.       I do not see the actual Table 1 in the pdf. I can only see a tag.

Minor comments

4.       It is still unclear what is “single-molecule super-resolution imaging” (Line 147, 172), especially since it is now specified to be not single-molecule localization microscopy, and not single-fluorophores imaging.

5.       Line 146. Ref 43 is a review on single-molecule localization microscopy, not single-molecule light microscopy.

6.       Line 144.  Should be live rather than fixed samples.

7.       Line 263. I’m glad the authors agreed with my previous minor comment 5. However, my concern was not that the text was too detailed, but that the statement of 1 fluorophore per DNA:RNA is not true.

8.       Line 704. Since molecular beacons are such a common tool, I believe is it worth being correct and precise here. The stem region that is standard for molecular beacons need to be complementary, there is no need for it to be palindromic at all. Do the authors have a source that suggest otherwise?

9.       Line 694. Molecular beacons used for RNA imaging are not RNA-based, they are typically DNA-based or uses modified nucleotides.
